# Cancer-Associated Adipocytes in Breast Cancer: Causes and Consequences

**DOI:** 10.3390/ijms22073775

**Published:** 2021-04-06

**Authors:** Ilona Rybinska, Nunzia Mangano, Elda Tagliabue, Tiziana Triulzi

**Affiliations:** Molecular Targeting Unit, Department of Research, Fondazione IRCCS Istituto Nazionale dei Tumori, 20133 Milan, Italy; ilona.rybinska@istitutotumori.mi.it (I.R.); nunzia.mangano@istitutotumori.mi.it (N.M.); elda.tagliabue@istitutotumori.mi.it (E.T.)

**Keywords:** cancer-associated adipocytes (CAA), adipogenesis, adipocyte dedifferentiation, signaling, breast cancer

## Abstract

Breast cancer progression is highly dependent on the heterotypic interaction between tumor cells and stromal cells of the tumor microenvironment. Cancer-associated adipocytes (CAAs) are emerging as breast cancer cell partners favoring proliferation, invasion, and metastasis. This article discussed the intersection between extracellular signals and the transcriptional cascade that regulates adipocyte differentiation in order to appreciate the molecular pathways that have been described to drive adipocyte dedifferentiation. Moreover, recent studies on the mechanisms through which CAAs affect the progression of breast cancer were reviewed, including adipokine regulation, metabolic reprogramming, extracellular matrix remodeling, and immune cell modulation. An in-depth understanding of the complex vicious cycle between CAAs and breast cancer cells is crucial for designing novel strategies for new therapeutic interventions.

## 1. Introduction

Breast cancer (BC) is the most common cancer in women and the leading cause of mortality for women with cancers worldwide [1]. Nowadays, it is widely accepted that BC progression is not only dependent on the intrinsic tumor characteristics but also on stromal cells (i.e., fibroblasts, endothelial, and various inflammatory cells and adipocytes) that constitute the tumor microenvironment (TME) [2]. The TME indeed actively contributes to the acquisition of cancer hallmark traits like angiogenesis, the epithelial-to-mesenchymal transition (EMT), proliferation, invasion, and metastasis [3].

Adipocytes are the main stromal cells in the breast, and research in recent decades has provided evidence that they are not just terminally differentiated cells impassive to the external environment. Indeed, besides differentiation, they can undergo dedifferentiation and trans-differentiation in many physiological processes, as well as in pathological conditions [4]. Cyclical dedifferentiation and re-differentiation processes of mammary gland adipocytes occur during reproduction [5]. Examples of adipocyte trans-differentiation are the formation of myofibroblasts from dedifferentiated dermal adipocytes and the browning of white adipocytes [6]. 

Nowadays, there is growing evidence that support the interaction between adipocytes and cancer cells, resulting in the involvement of adipocytes in all phases of BC progression (reviewed in [7]). Epidemiological studies reporting an association between obesity and the higher incidence/progression of BC have sustained the role of adipocytes in BC progression [8,9]. The study of this crosstalk in the breast is interesting because, from the first steps of cancer initiation, mammary tumors are located next to the adipose tissue. The intimate crosstalk between cancer cells and adipocytes induces their dedifferentiation in terms of a reduction of terminal differentiation with a reduction and increase in the expression of differentiation markers and several pro-tumoral molecules, respectively. Due to their contribution to tumor cell aggressiveness, tumor-modified adipocytes have been named cancer-associated adipocytes (CAAs) [10]. Several mechanisms underlying CAA-driven cancer progression have been proposed in analogy to features of adipocytes in obesity (reviewed in [11]), and although some candidate molecules secreted by tumor cells have been proposed to trigger the process of adipocyte dedifferentiation, the fundamental cellular and molecular mechanisms of this intricate interaction have not been completely elucidated. 

This article reviewed the recent studies on the mechanisms underlying the complex bidirectional interaction existing between CAAs and BC cells. The role of tumor-secreted molecules in this interaction will be discussed with a focus on pathways already described to be relevant in the adipogenesis process. 

## 2. CAA Characterization

CAAs have been described for the first time upon co-culture of 3T3-F442A mature adipocytes with BC cells in vitro [10]. Adipocytes derived from the differentiation of murine 3T3-F442A or 3T3-L1 cells, that are clonal sublines isolated from 3T3 mouse embryonic fibroblasts, undergo sequential phenotypic and functional alterations that differentiate them from the mature adipocytes from which they derive upon exposure to cancer cells or their conditioned media in vitro. They first decrease their size, lipid content, and expression of adipocyte differentiation markers such as resistin, adiponectin, and fatty acid binding protein (FABP4, also known as adipocyte protein 2, aP2), and then decrease their transcriptional regulators, the peroxisome proliferator-activated receptor γ (PPARγ) and co-activator CCAAT/enhancer binding protein α (C/EBPα), leading to irregular shapes and small/dispersed lipid droplets [10,12] that make them similar to brown adipocytes [11]. Accordingly, a higher expression of the uncoupling protein 1 (UCP1) in CAAs has been described [13]. CAAs re-express preadipocyte marker genes and gain proliferative capacity [14]. Moreover, they present an activated phenotype characterized by the overexpression of chemokines (i.e., CCL2 and CCL5), inflammatory cytokines (i.e., interleukin (IL)1β, IL-6, tumor necrosis factor-alpha (TNFα)), and proteases (MMP11) [10,14]. Next, they reorganize their actin cytoskeleton, increase fibroblast-like biomarkers such as fibroblast activation protein a (FAP), chondroitin sulfate proteoglycan, and smooth muscle actin (a-SMA), and acquire a fibroblast-like morphology [14]. In terms of metabolic changes, CAAs increase several catabolic processes, releasing metabolites, such as lactate, pyruvate, free fatty acids (FFAs), and ketone bodies [15]. FFAs derive from adipocyte lipolysis induced by the activation of the hormone-sensitive lipase (HSL) by the BC-conditioned medium (CM) [16]. While HSL phosphorylation is indicative of activated lipolysis, evidence coming from a well-recognized glycerol assay (in cancer-adipocyte co-culture/CM settings) remains controversial, because lipolysis is not restricted to adipocytes, it can be employed by cancer cells [17], and it can even be reinforced in the presence of adipocytes [16]. 

A CAA-activated phenotype has been confirmed in a co-culture with tumor cells from several histotypes (prostate, pancreas, melanoma, lung, ovary, hepatocarcinoma, and leukemia [18]) and in vivo at the invasive front of human BCs [10,19,20]. Indeed, adipocytes at the invasive tumor front are smaller and express CAA-related molecules compared to adipocytes distant from the tumor or derived from healthy donors. Moreover, at the tumor center, adipocytes are rarely found, whereas a high number of fibroblast-like cells are present in the TME, suggesting that they are derived from adipocyte dedifferentiation. Even if a specific marker of adipocyte-derived fibroblasts (ADFs) has not already been found, Bochet and coworkers demonstrated that these cells accumulate in the murine mammary carcinoma tissue and contribute to the desmoplastic reaction found around the tumor [14]. The possibility that fibroblasts derive from adipocytes is supported by lineage-tracing studies in animal models both in physiologic (e.g. lactating mammary gland [21]) and pathologic (fibrotic dermis [22] and liposarcoma [23]) conditions.

## 3. Regulation of Adipocyte Differentiation

Adipogenesis is the process through which mesenchymal stem cells (MSC) upon adipogenic stimuli differentiate into adipocytes. It is a complex and highly orchestrated program involving both extracellular signals and a complex network of transcription factors and cofactors. PPARγ, a member of the nuclear receptor superfamily, is the master regulator of adipogenesis, and it is indeed necessary and sufficient to induce and maintain adipocyte differentiation [24]. Moreover, most adipogenic and repressing factors function at least in part by activating and inhibiting, respectively, its expression/activity [25,26]. Another crucial transcription factor, with which PPARγ creates a positive feedback regulating loop, is C/EBPα (Figure 1). The product of the cooperation of these two transcription factors is the expression of genes involved in insulin sensitivity, lipogenesis, and lipolysis in differentiating and mature adipocytes, including the glucose transporter GLUT4, FABP4, adipose triglyceride lipase (ATGL), lipoprotein lipase (LPL), perilipin, a lipid droplet coating protein, and the secreted factors adiponectin and leptin [27]. 

This highly regulated transcriptional program is activated downstream of the signaling pathways that transduce information from intracellular and extracellular factors (Figure 2). Wnt, Hedgehog, Notch, and transforming growth factor (TGF) β pathways with a key role in embryogenesis have emerged as crucial modulators of cell differentiation processes, like adipogenesis. In addition, several new mechanisms regulating transcription factors implicated in adipogenesis have also been described. Among them, histone acetyltransferase (HATs), deacetylase (HDACs), methyltransferases, and demethylase were described to be involved in the dynamic remodeling of chromatin and in adipogenic gene expression (reviewed in [28]). In addition, an increasing number of miRNAs has been identified to regulate key transcription factors involved in adipogenesis, providing an additional level in the regulation of multiple genes at once [29]. A redox-state was also found relevant for the 3T3-L1 mitotic clonal expansion (MCE) phase and terminal differentiation modulating C/EBPβ DNA binding activity [30].

### 3.1. Wnt Signaling 

An extensive number of studies in the literature have demonstrated the importance of Wnt signaling in adipocyte development, both in vitro and in vivo, and have shown that the suppression of this pathway is essential for adipogenesis to proceed (reviewed in [31]). Wnts are a highly conserved family of autocrine and paracrine ligands known to have key roles in cell fate and development (reviewed in [32]). The binding of Wnt proteins to the frizzled receptors and the low-density lipoprotein receptor-related protein 5 (LRP5) and LRP6 in 3T3-L1 and 3T3-F442A preadipocytes induces β-catenin translocation to the nucleus where it interacts with the TCF/LEF transcription factors to block the expression of PPARγ and C/EBPα [33]. In particular, the inhibition of WNT10b in 3T3-L1 preadipocytes, which is exerted by cAMP agonists contained in the adipogenesis cocktail used in vitro to differentiate preadipocytes, is required for adipocyte differentiation [34]. Wnt ligands can also signal through β-catenin-independent pathways, known as noncanonical signaling [32]. WNT5A and B, two Wnt proteins responsible for turning on noncanonical signaling, have pro-adipogenic properties in 3T3-L1 cells [35]. WNT5A antagonizes the WNT/β-catenin pathway by GSK3β-independent β-catenin degradation [36], and WNT5B by preventing the translocation of β-catenin in the nucleus induced by the canonical Wnt ligands [37]. However, their effects are not always consistent and in vivo experiments of the role of noncanonical Wnts in adipogenesis have generated conflicting results [31]. It is possible that different noncanonical pathways (e.g., small Rho GTPases, JNK/AP-1, Ca^2+^/NFAT) exert opposing effects on adipogenesis, providing complexity to the system.

### 3.2. Hedgehog Signaling

The Hedgehog (Hh) pathway is activated by the receptor Patched (PTC) and its ligands Sonic Hedgehog (SHH), Indian Hedgehog (IHH), or Desert Hedgehog (DHH). Downstream, the signal ends with a translocation of the Gli2/3 transcription factor to the nucleus (reviewed in [38]). A basal Hh signaling activity was found in 3T3-L1 cells; this activity decreases as soon as cells differentiate into adipocytes [39]. Accordingly, the anti-adipogenic properties of Hedgehog signaling have been clearly evidenced in vitro on 3T3-L1 and other murine preadipocytes, even if the molecular bases are not completely understood [38]. The induction by SHH of antiadipogenic transcription factors such as GATA2 that directly suppress PPARγ [40] has been proposed [39]. Moreover, it is not clear if it impacts adipogenesis directly or indirectly through the upmodulation of Wnt proteins. In addition, a novel crosstalk between these two pathways has been proposed, in which Gli3R, generated in the absence of Hh signaling, linked and antagonized the active forms of the Wnt canonical transcriptional effector, beta-catenin [41], potentially influencing adipogenesis. 

### 3.3. Notch Signaling 

Notch signaling is an evolutionarily conserved pathway that plays essential roles in cell fate choice, organ development, metabolism, and tumorigenesis (reviewed in [42]). Notch is a transmembrane receptor for membrane-bound ligands of the Delta and Serrate/Jagged families. The binding of ligands to Notch induces the proteolytic cleavage of Notch by the γ-secretases and translocation of its intracellular domain (N1ICD) to the nucleus. In the nucleus, N1ICD binds to and activates different transcription factors [42]. While the data obtained with the Notch inhibitor (i.e., γ-secretases inhibitor) in vitro in murine and human primary adipose-derived stem cells together showed that Notch1 is a negative regulator of adipogenesis, in the 3T3-L1 cell line, Notch signaling has been shown to either promote or inhibit adipogenic differentiation (reviewed in [43]). The Notch suppression or loss of HES1, with one transcriptional factor mediating some Notch effects, blocks the differentiation of 3T3-L1 cells and induces the expression of Pref-1 (reviewed in [44]). However, other studies in these cells have shown that adipocyte differentiation is blocked when Notch signaling is activated by Jagged or by constitutively active HES1 [45]. Given these contradictory reports, the role of Notch signaling in regulating the differentiation of the adipocyte progenitor cells need further studies.

### 3.4. TGFβ and BMP Signaling

TGFβ superfamily signaling regulates many cellular processes like apoptosis, inflammation, fibrosis, and adipocyte differentiation (reviewed in [46]). Two types of ligands exist: TGFβs and BMPs, which bind to two different receptors activating SMAD-dependent pathways (TGFβ/SMAD signaling) or SMAD-independent pathways, which mainly promote ERK, JNK, and p38 MAPK kinase signaling. Upon phosphorylation of the receptor, SMAD proteins form a dimer with SMAD4 and translocate to the nucleus, where they regulate the transcription of target genes. It is now established that TGFβ/SMAD signaling has a dual role in the adipocyte differentiation process (reviewed in [47]). BMP signaling promotes adipocyte maturation through the SMAD1-dependent mechanism, while TGFβ signaling inhibits it through SMAD2/3-dependent inhibition of the C/EBPs transcriptional activity in NIH3T3 cells [47,48]. Indeed, BMP2 was described to induce adipocyte differentiation in 3T3-L1 preadipocytes and C3H10T1/2 murine MSCs both by SMAD and p38 kinase signaling [49]. Then, Schnurri 2 (SHN2) was found to control BMP-dependent adipogenesis in mice and in vitro interacting in the nucleus with SMAD1/4, inducing the expressions of PPARγ and adipogenesis [50]. On the contrary, a blockade of TGFβ signaling by the expression of a dominant negative receptor in 3T3-F442A cells, or by the inhibition of SMAD3, induced adipogenesis [51]. Recently, TGFβ/SMAD signaling was also found to be crucial in modulating the adipocyte commitment of MSCs (reviewed in [52]).

### 3.5. Insulin Signaling

Insulin is a hormone involved in the differentiation of adipocytes other than their metabolic features through the activation of an intracellular signaling cascade involving the insulin receptor (IR), insulin receptor substrate (IRS) proteins, phosphoinositol 3-kinase (PI3K), and protein kinase B (AKT) (reviewed in [53]). Downstream components of the insulin/IGF1 signaling cascade are crucially important for adipogenesis. Indeed, the loss of IRS (mainly IRS1) and/or the inhibition of PI3K in embryonic murine fibroblasts, and the loss of AKT inhibit adipogenesis in vitro and in vivo [54,55]. Another downstream effector of insulin action, through the canonical PI3K-AKT signaling pathway, is the mammalian target of rapamycin (mTOR). mTORC1 signaling has been demonstrated to be essential for white adipocyte differentiation being implicated in promoting both the lineage commitment, clonal expansion, and terminal differentiation of preadipocytes to mature adipocytes through several effectors (reviewed in [56]). Available evidence has indicated that the mTORC1 signaling pathway may also regulate lipid metabolism via the inhibition of relevant molecules implicated in lipolysis (i.e., HSL, LPL, and ATGL) [56]. IRS signaling also promotes the phosphorylation of CREB, which is important for adipogenesis. In addition, AKT activation leads to the nuclear exclusion of the anti-adipogenic transcription factors FOXO1, FOXA2, and GATA2, allowing adipocyte differentiation to proceed [57].

### 3.6. The MAPK Pathway

The mitogen-activated protein kinase (MAPK) pathway plays a pivotal role in several cellular processes such as proliferation and differentiation through three groups of MAPKs: the extracellular signal-regulated kinases (ERKs); the stress-activated protein kinases that are activated by stress stimuli c-Jun amino-terminal kinases (JNKs); and p38 MAPK (p38) (reviewed in [58]). A fine-tuning of the MAPKs regulates both normal and pathological adipogenesis (reviewed in [59]). Indeed, ERK was described to be necessary for the 3T3-L1 mitotic clonal expansion, a proliferative step that is initiated by insulin, which is known to activate the ERK pathway. Then, this signal transduction pathway needs to be shut-off to avoid PPARγ phosphorylation and to proceed with adipocyte maturation [59,60]. Regarding p38 kinases, both the induction and inhibition of adipogenesis were demonstrated in different studies, making their precise role in adipogenesis still unclear (reviewed in [61]). It is likely that p38 has a different role according to the differentiation stage, similarly to ERK. Moreover, recent findings using phosphoproteomics to monitor the alteration of nuclear proteins during the early stages of 3T3-L1 differentiation and studying the interaction between BMP signaling and p38 suggest an adipogenic role of the p38 family through its downstream substrates ATF2 and CREB [49,62]. ERK and JNK can also regulate adipogenesis other than insulin resistance through the inhibitory phosphorylation in the serine residues of IRS1 [63]. 

### 3.7. FGFs

The FGF family comprises 22 members with diverse functions in development, metabolism, and neuronal activities. FGF10 plays unique roles in adipocyte development, stimulating both preadipocyte proliferation and adipogenesis through the Ras/MAPK pathway [26]. The positive role in early adipogenesis has been described mainly in 3T3-L1 cells also for FGF2 and for FGF1 [26]. Moreover, recent studies in in vivo and ex vivo murine models indicated that FGF1 is also released by mature adipocytes to induce adipocyte precursor differentiation as a central feedback mechanism by which mature adipocytes control adipogenesis during the development of obesity [64]. 

### 3.8. Inflammatory Molecules

Cytokines and pro-inflammatory cytokines secreted both by adipocytes and immune cells infiltrating the adipose tissue were described to be potent regulators of adipogenesis (reviewed in [65]). TNFα is considered an inhibitor of adipogenesis through multiple mechanisms. Indeed, studies in 3T3-L1 and 3T3-F442A cells showed that through its receptor TNFR1, it activates NF-kB and MAPK inhibitory signaling (JNK and ERK) other than the Wnt/β-catenin inhibitory pathway, resulting in both the inhibition of PPARγ at the transcriptional level and the reduction of its nuclear localization [66,67,68]. Other pro-inflammatory molecules demonstrated to negatively affect the adipogenesis of human adipose-derived MSCs or preadipocytes are IL-1α and IL-1β that mainly exerts their function through the activation of the NF-kB pathway [69,70]. 

IL-6, a pro-inflammatory member of the Gp130 cytokine, was described to induce insulin resistance and to inhibit adipogenesis in human preadipocytes [71]. Its activity is likely to be mediated by the activation of the Janus-activated kinase/signal transducer and activator of transcription factor (JAK/STAT) molecules, which then decrease the PPARγ and CEBP/α expressions, reducing the adipogenesis rate (reviewed in [72]).

Moreover, mechanistic studies in 3T3-L1 cells demonstrated a role for IL-15, IL-4, and IL-17 in inhibiting adipogenesis through the upregulation of calcium-dependent mechanisms, the downregulation of PPARγ in a STAT6-dependent manner, or through activation of antiadipogenic Krüppel-like family (KLF) members (KLF2 and KLF3), respectively [73,74,75]. Interferons (IFNα and IFNγ) were demonstrated to reduce adipocyte differentiation through the activation of the JAK/STAT pathway [76] or inhibition of the Hh pathway [77].

## 4. Regulation of CAA Induction

CAA occurrence is the result of the adipocyte dedifferentiation, delipidation, and acquisition of an activated phenotype. Whether these processes are all induced by a single molecule released by tumor cells or are differently regulated still need to be addressed. Indeed, only a few studies have so far demonstrated a mechanism at the bases of CAA induction in cancer and other pathologies (Figure 3), focusing on molecules already described to have a role in blocking adipogenesis processes. 

### 4.1. The Wnt Pathway in the Crosstalk between CAA and Cancer Cells

The reactivation of the Wnt signaling has been shown to be implicated in adipocyte dedifferentiation, consistent with its role as a main inhibitor of adipogenesis. Indeed, the WNT3A released by murine and human BC cells has been identified to be sufficient for the induction of 3T3-F442A adipocyte dedifferentiation [14,78]. WNT3A treatment induced the upmodulation of WNT10b in adipocytes and several typical undifferentiated cell markers, including Pref-1 and GATA2, and the blockade of this pathway partially reversed the adipocyte-derived fibroblast phenotype [14]. Another study suggested that 3T3-L1 adipocyte dedifferentiation occurs through the Wnt pathway, the non-canonical one, by the release of WNT5A from human MiaPaCa2 pancreatic cancer cells [79]. In detail, upon the co-culture with adipocytes, tumor cells increased early the expression of WNT5A that activates c-Jun and AP1 in adipocytes. Both a WNT5A neutralizing antibody and recombinant SFRP5, a competitive inhibitor of the WNT5A receptor, reverted the activation of PPARγ and c/EBPα in adipocytes, blocking their reprogramming toward fibroblast-like cells. It is to acknowledge that some controversial issues regarding the role of tumor-released WNT5A in mediating CAAs have been highlighted, mainly based on the described ability of WNT5A to induce adipogenesis and on the expression of this ligand also in adipocytes [80]. 

Other evidence to support the role of the canonical Wnt-β-catenin pathway in adipocyte dedifferentiation is the involvement of this pathway in the TNFα-induced [78] and in the compression-induced adipocyte dedifferentiation (compression-induced dedifferentiated adipocytes, CiDAs), resulting from various types of physical stresses like osmotic compression on adipocytes [81]. While the inhibition of Wnt signaling significantly reduced the dedifferentiation rate, partially restoring lipid droplet accumulation in adipocytes, the rescue of the signaling via the addition of the exogenous WNT3A ligand recovered the CiDAs phenotype [81]. Interestingly, WNT3A expression in human BC cells was found to be dependent on the activation of focal adhesion kinase (FAK) [82] that mediates the intracellular transduction of mechanical signals, like those occurring in stiff tissues [83], supporting the role of physical stress also in the tumor-dependent induction of CAAs.

The need of the modulation of the Wnt pathway in adipocytes to obtain CAAs is in line with the described interaction between the Wnt pathway and PPARγ [84], whose activation has to be switched off to make dedifferentiation occur. PPARγ and β-catenin signaling in adipose cells seem to be mutually antagonistic: PPARγ has been shown to induce β-catenin degradation through direct association and/or indirectly through transcriptional regulation. However, it is important to note that further steps in addition to PPARγ repression are needed to obtain a full adipocyte reprogramming typical of CAAs. Indeed, PPARγ depletion in 3T3-L1 adipocytes was not sufficient to induce a marked adipocyte dedifferentiation, because some genes like GATA2 were not reversed by its knockdown, remaining low [85]. The forced expression of GATA2 in 3T3-L1 adipocytes depleted for PPARγ led to a more preadipocyte-like gene expression profile. In this context, the TNFα activity on 3T3-L1 adipocytes is limited to PPARγ repression and β-catenin stabilization, resulting in a rapid loss of cellular lipids and eventually apoptosis [78], but only WNT3A allows the cell to assume an immature phenotype, rendering them able to assume a myofibroblast morphology and an activated phenotype expressing Pref-1, GATA2, and WNT10b. Again, this action seems to be mediated by other effects like the activation of ERK signals, supporting the complex regulatory network behind CAA induction. 

### 4.2. Other Potential Modulators of Adipocyte Reprogramming in Cancer

Other tumor-derived players in CAA induction have been recently proposed, even if their mechanisms of action are not completely understood. Exosomes from human hepatocellular carcinoma HepG2 cells were found to be internalized by adipocytes derived from human adipose-derived MSCs in which they induced an inflammatory profile similar to that described in CAAs [86]. Although the implicated molecules have not been identified, the activation of AKT, STAT5α, GSK3β, ERK1/2, and NF-kB pathways in adipocytes treated with tumor-derived exosomes, as analyzed by phosphoproteomics and Western blot, was demonstrated, supporting the relevance of these pathways and the exosome method of cell–cell interaction in the dedifferentiation process [86]. Accordingly, extracellular vesicles (EVs) derived from Lewis lung carcinoma (LLC) cells induce lipolysis in 3T3-L1 adipocytes through their IL-6 content, which activates the JAK/STAT3 pathways in adipocytes [87]. In addition, exosomal miR-144 and miR-126, highly secreted from human BC cells upon co-culture with 3T3-L1 adipocytes, were described to promote adipocyte reprogramming, reducing PPARγ expression and disrupting the IRS signaling, respectively [88]. In another model, PPARγ was found also downregulated by miR-155 secreted by BC cells participating in the metabolic remodeling of 3T3-L1 adipocytes [89]. In addition, it has also been recently evidenced that adrenomedullin (ADM) secreted by human BC mammospheres promotes the increase in UCP1 expression in human adipocytes differentiated from breast adipose-derived MSCs and augments HSL phosphorylation through the p38-MAPK, leading to delipidation in adipocytes [90]. 

### 4.3. Other Molecular Pathways Involved in Adipocyte Dedifferentiation

Other pathways potentially involved in the tumor-induced adipocyte dedifferentiation are the Notch and TGFβ pathways, described to have a role in adipocyte reprogramming in liposarcoma and fibrosis development [22,23]. The over-expression of Notch ligands in human BC has been shown to correlate with poor prognoses [91], and TGFβ production was demonstrated to be relevant in BC progression in several studies (reviewed in [92]).

The analysis of Ad/N1ICD mice that overexpress the activated form of the Notch1 receptor (N1ICD) specifically in mature adipocytes showed that they have a significantly lower body fat mass compared to the controls, and gene expression analysis in the adipose tissue depots showed a significant repression of lipogenic and adipogenic pathways (C/EBPα, C/EBPβ, and SREBF1) [93]. The liposarcoma that developed in this mouse model is accompanied by reduced expression levels of adipogenic markers and increased expression levels of preadipocyte markers and oncogenes such as Pref-1 and MDM2, supporting the dedifferentiated status of the Notch-activated adipocytes. As PPARγ was overexpressed in liposarcoma cells, it is likely that the dedifferentiation in Ad/N1ICD adipocytes was derived from the deficient activation of PPARγ due to the suppression in the cells of lipid metabolism pathways that supply ligands to PPARy. Accordingly, synthetic PPARγ ligand supplementation induces the re-differentiation of Ad/N1ICD adipocytes and prevents liposarcoma development in Ad/N1ICD mice, supporting Notch activation as sufficient to induce the dedifferentiation and tumorigenic transformation of mature adipocytes [23]. 

TGFβ1 has also been shown to modulate the process of adipocyte dedifferentiation into fibroblast-like cells in the pathogenesis of cutaneous fibrosis [22]. In a mouse model of bleomycin-induced fibrosis, intradermal adipose tissue loss was accompanied by the early downregulation of PPARγ and other adipocyte-associated genes, followed by the expression of myofibroblast-associated genes. Tracing experiments in the AdipoP-Cre–tdTomato–transgenic mice demonstrated the adipocyte origin of myofibroblast cells infiltrating the lesional dermis. Ex vivo studies with human subcutaneous adipocytes indicated that TGFβ1 treatment for 24 h is responsible for a large-scale transcriptional reprogramming, with decreasing expression of many adipogenic genes and an increase in the expression of pivotal fibrogenic genes. The upmodulation of WNT5A and SFRP2 upon treatment with TGFβ1 supports that it may have an effect on other pathways to co-regulate dedifferentiation [22]. As adipose tissue loss is also frequently associated with fibrosis in other disorders (e.g., autoimmune diseases, scarring alopecia, anorexia, and cancer cachexia), it is likely that adipocyte dedifferentiation is at the basis of these pathologies too. Mouse models of these disorders were generated through PPARγ deletion or Wnt and TGFβ overexpression specifically in adipocytes, supporting the causal involvement of these signaling pathways in the progressive disappearance of intradermal adipocytes and their replacement with fibrotic tissue in several disorders [94,95]. 

TGFβ also plays a major role in adipose tissue remodeling through the induction of ECM proteins, such as collagens, whose composition and dynamics are of crucial importance in adipogenesis [96]. Even if the demonstration of the causal impact of TGFβ/ECM modulation during dedifferentiation is still necessary, the described upmodulation of collagen I and VI genes by TGFβ1 in human dedifferentiated fat (DFAT) cells in vitro supported that this growth factor could modulate adipocyte differentiation and dedifferentiation through the modulation of ECM. In this context, MMP11, also called stromelysin 3, whose substrate is the native alpha3 chain of collagen VI [97], has been reported as a negative regulator of adipogenesis and as an inductor of adipocyte dedifferentiation [20]. Even if further studies are necessary to understand whether MMP11 induces the lipolysis of mature adipocytes or only reduces preadipocyte differentiation, it is possible that ECM components and their bioactive fragments [98] signal inside adipocytes through integrins participating in CAA generation. Another molecule that was suggested to participate in adipocyte dedifferentiation is DPP4 (also known as CD26), a transmembrane serine peptidase of the prolyl peptidase family involved in ECM degradation by cleaving collagens [99]. The authors demonstrated that the addition of a DPP4 inhibitor during the ceiling culture of human subcutaneous mature adipocytes, described to generate DFAT cells, modestly reduced C/EBPα expression in a dose-dependent manner, supporting a potential role for this enzyme in adipose tissue remodeling and cell plasticity.

## 5. CAA-Derived Molecules in Cancer Progression

The modification of the adipocyte phenotype by cancer cells induces a strong effect in tumor cells and possibly in other cells of the TME. In vitro and in vivo experiments demonstrated that CAAs play essential roles in favoring the proliferation, angiogenesis, dissemination, invasion, and metastasis of BC (reviewed in [18,100]). The wide range of molecules and metabolites secreted by CAAs can mediate these effects directly or indirectly, through the induction of the tumor immunosuppressive microenvironment (Figure 4). Uncovering the functions and the mechanisms of CAA-derived molecules in BC progression are essential to the establishment of potential therapeutic interventions. 

### 5.1. Adipokines

Adipokines are cytokines secreted by adipocytes with autocrine and paracrine functions. Compared with normal fat cells, CAAs are characterized by a dysfunctional, vicious phenotype and a more aggressive secretome. Here, we especially focus on adipokines such as leptin, adiponectin, IL-6, and autotoxin (ATX), whose impaired secretions from CAAs promoted the cancer aggressive phenotype. 

Leptin is a hormone and satiety factor produced mainly by adipocytes known to regulate several processes associated with cancer progression, such as immune response, angiogenesis, and proliferation (reviewed in [101]). Its elevated serum levels were positively correlated with BC risk, aggressiveness, and bad patient prognosis [102]. Furthermore, expression levels of leptin and its receptor (ObR) in primary and metastatic BCs were found to be increased, relatively to noncancer tissues [103]. Among others, leptin can regulate the expression of cyclin D, p53, and apoptosis; thus, it is implicated in breast carcinogenesis [101]. In vitro, it was shown to enhance BC proliferation, invasion, and migration, and it was shown to promote CSCs enrichment and the EMT phenotype [104], but the spectrum of its action is presumably much wider, as several cells in TME possess its receptor. 

Adiponectin is one of the most widely studied adipokines to date and it is a recognized marker distinguishing fat cells from other cell types (reviewed in [105]). Adiponectin has a plethora of effects on many different target tissues, and it is widely regarded as an anti-apoptotic, anti-inflammatory, anti-fibrotic, and insulin-sensitizing agent. Adiponectin is recognized as an anticancer agent, also in BC, and its levels are lower in CAAs in vitro and in BC-associated fat compared to fat adjacent to benign lesions [13]. Adiponectin binds to adiponectin receptors (AdipoR1 and AdipoR2), expressed in normal and malignant BC tissues and through the activation of AMPK, and PI3K/AKT inhibition was shown to suppress the growth and migration of BC cells [7]. Although, several epidemiological studies associated low circulating levels of adiponectin with an increased BC risk and more aggressive phenotype, in particular, in post-menopausal women (regardless of BMI), this relationship has been observed prevalently in ER/PR-negative BCs [106]. Accordingly, the dichotomic adiponectin effects on BC growth and progression in regard to ERα status were documented [106]. Several reports supported a protective role of adiponectin also in ERα-positive BC, but in this particular BC subtype, divergent actions of the adipokines have been observed. 

IL-6 secretion is an inextricable element of the CAAs phenotype. The increased secretion of IL-6 by human adipocytes differentiated in vitro was shown to promote tumorigenesis by the activation of STAT3, and the IL-6/STAT3 axis has been identified as one of the principal mechanisms by which adipocytes induce the EMT-phenotype in BCs [107]. Furthermore, it was shown that IL-6 circulating levels were associated with aggressive features and can worsen prognosis in BC patients [108].

Autotoxin (ATX), a secreted lysophospholipase D that generates the bioactive lipid mediator lysophosphatidate (LPA) from lysophosphatidylcholine (LPC), is mainly produced by adipocytes (reviewed in [109]). One of the main functions of ATX and LPA is in wound healing, in which LPA facilitates wound repair by augmenting the cell proliferation, migration, and survival of fibroblasts in the damaged areas. In cancer, a wound that never heals, it also enables immune evasion and stimulates angiogenesis. Moreover, in BCs, elevated levels of ATX-LPA are associated with resistance to chemotherapeutics and radiotherapy [109]. 

In addition to the adipokines mentioned above, there are multiple growth factors, cytokines, and chemokines produced by CAAs [110], which are implicated in a vicious cycle between adipocytes and BC cells in the TME, inducing proliferation and metastasis, such as visfatin and resistin [111], CCL2 and CCL5, insulin-growth factor 1 (IGF-1) [112], hepatocyte growth factor (HGF), insulin-like growth factor binding protein 2 (IGFBP-2) [113], TGFβ, vascular endothelial growth factor (VEGF), TNFα, as well as granulocyte colony-stimulating factor (G-CSF) [114]. 

### 5.2. Metabolic Reprogramming

The adaptive mechanism of metabolic reprogramming is one of the hallmarks of cancer, as it is not only needed to fuel energy demands or to support biomass generation, but it also has essential roles in other cancer features such as migration, invasion, and metastasis [115]. In the TME, tumor cells behave as parasites sequestering metabolic substrates, including lactate, glutamine and FAs, from stromal cells via the stimulation of catabolic pathways, such as autophagy, glycolysis, and lipolysis [116]. These metabolites are used as substrates for anabolic metabolism in cancer cells.

During the process of CAAs generation, the HSL- and ATGL-mediated lipolysis of triglycerides (TGs) induce the release of FFAs and the CAA-delipidated morphology (reviewed in [15]). The uptake of FFAs in cancer cells is mediated by several proteins, including fatty acid translocase (FAT)/CD36, fatty acid transport proteins (FATPs)/SLC27A, low-density lipoprotein receptor (LDLR), and fatty acid binding proteins (FABPs) (reviewed in [117]). The uptake of FFAs by CD36 was described to increase BC cell growth in vitro [118], and its expression in human BC samples was associated with poor prognoses [119]. FFAs transferred to tumor cells might promote tumor progression through various mechanisms such as (i) metabolic remodeling, (ii) epigenetic changes, (iii) elevated reactive oxygen species (ROS) production and the consequent acquisition of more aggressive tumor traits, as well as iv) prompt transcriptional regulation involved in tumor progression [15]. Additionally, recent studies have indicated that lipid transfer from adipocytes to cancer cells can also occur through extracellular vesicles (EVs) in a process independent from, but possibly reinforced by lipolysis [120]. That discovery translates not only to the modality of lipid transport, but also to their fate in cancer cells. As EVs, lipid cargo is very complex, and lipid species that transfer from adipocytes to tumor cells in such a way (sphingomyelin, cholesterol, lysophosphatidylcholine, and eicosanoids) are much more heterogeneous when compared to simple lipolysis, leading mainly to palmitic, oleic, and linoleic acid release [15]. Curiously, the characteristic feature of exosomes derived from adipocytes is that they also carry proteins mainly implicated in fatty acid oxidation (FAO) [120], and their transfer together with substrates necessary for FAO (i.e., lipids) were shown to exacerbate melanoma cells migration [120,121]. The nutrient transfer between adipocytes and cancer cells besides fats also refers to amino acids such as glutamine, which is of great importance in tumor cell metabolism, enabling the reduction of NADP+ to NADPH required as an electron donor in lipid synthesis, nucleotide metabolism, as well as the maintenance of reduced glutathione (GSH) [122]. 

While most cancers metabolize glucose even when oxygen is available (“Warburg effect”), their metabolism changes upon interaction with stromal cells. Indeed, a tumor-oriented stroma undergoes aerobic glycolysis (“reverse Warburg effect”), and the resulting byproducts are shuttled to the cancer cells to support mitochondrial oxidative phosphorylation (OXPHOS) [115]. Monocarboxylate transporters (MCTs) participate in the transport of lactate and pyruvate and are associated with poor prognosis in BCs. They are at the core of metabolic symbiosis between hypoxic and normoxic cancer cells, but also play important roles in the dynamic monocarboxylate shuttle between cancer cells and fibroblasts, as well as adipocytes [123]. Indeed, the high expressions of MCT1 (mediating lactate uptake) and MCT4 (mediating lactate efflux) in tumor tissues were associated with poor patient outcomes, and this association was reinforced combining the expression of tumor MCT1 with MCT4 overexpression in adjacent AT [124]. 

In addition, recent in vitro studies have shown that the interaction between 3T3-L1 mature adipocytes and cancer cells potentiates both ketogenesis in adipocytes and ketolytic activity in BC cells; such a metabolic adaptation was further confirmed in BC tissues located near stromal adipocytes [88]. β-hydroxybutyrate (BHB) generated from ketogenesis and released by adipocytes is more than just a metabolite, promoting malignancy also through the epigenetic upregulation of tumor-promoting genes like IL-1β and lipocalin 2 [125]. The induction of a ketone-specific gene signature was associated with poor clinical outcomes in human BC patients [126].

### 5.3. ECM Remodeling

The TME is composed of stromal cells and a noncellular component, termed the ECM, which is a meshwork of polymeric proteins and accessory molecules providing both a biochemical and biomechanical context [83]. Cancer progression and invasion requires ECM remodeling. A variety of molecules produced by adipocytes, including multiple ECM components and regulating enzymes, impact tumor ECM mechanical support. 

CAAs overexpress collagen VI and MMP11 that can cleave this collagen chain, producing endotrophin [20,127]. Collagen VI promotes the growth and survival of BC lesions by signaling through NG2/chondroitin sulfate proteoglycan receptors in cancer cells [127], and endotrophin contributes significantly to tumor growth, angiogenesis, fibrosis, and EMT, as demonstrated in vitro and in vivo [128]. In addition, recent research has evidenced that CAAs could impact BC progression, also remodeling collagen I assembly [129]. In detail, tumor-derived plasminogen activator inhibitor 1 (PAI-1), through the induction of the PI3K/AKT pathway in CAAs, activates the expression of procollagen-lysine 2-oxoglutarate 5-dioxygenase 2 (PLOD2) that induces the alignment of collagen I fibers, further promoting BC metastasis. The relevance of PAI-1 and PLOD2 in BC-CAA crosstalk was confirmed in clinical specimens of BC patients and in a 3D collagen invasion model [129]. 

CAAs were also described as able to regulate ECM remodeling, modulating the expression of metalloproteases in tumor cells. The short-term co-culturing of cancer cells with adipocytes was enough to induce the upregulation of MMP2 in MCF7 cells, increasing their invasiveness [113]. MMP2 and MMP9 production by tumor cells were described to also be controlled by adipocyte-derived leptin and IL-6 through the activation of the FAK- and Src-dependent pathways [130]. 

### 5.4. Immune Cell Modulation by CAA-Released Molecules

The tumor immune microenvironment (TIME) includes myeloid-derived suppressor cells (MDSCs), tumor-associated macrophages (TAMs), natural killer cells (NKs), dendritic cells (DCs), and T cells [131]. The list of molecules released by CAAs with a potential role in recruitment and function of immune cells includes: adiponectin, leptin, CCL2, CCL5, IL-1β, IL-6, IL-8, TNFα, VEGF, PAI-1, SFRP5, serum amyloid A (SAA), and hyaluronan (reviewed in [132,133]). Additionally, as TME is metabolically restrictive for infiltrating immune cells, metabolic products like FAs and lactate derived from CAAs could profoundly impact innate and adaptive immune cell homeostasis and differentiation [134], ultimately driving immune escape and tumor progression.

A prominent example of the importance of FAs in immunometabolism is that of neutrophils—the most abundant, short-lived mediators of innate immunity—which, requiring FAs for proper differentiation, could exploit the interplay with CAAs [135]. In addition to FFAs, CAA could promote neutrophil differentiation, increasing the glycolytic flux in neutrophils and potentially promoting the function of tumor-associated neutrophils (TANs). Neutrophils are recruited in the TME by CAA-derived IL-8, and this process was shown to be a part of the metastatic cascade in BC [136]. In pancreatic ductal adenocarcinoma as well, the recruitment of neutrophils by CAA-secreted IL-1β is relevant in tumor progression [132]. As adenosine derived from ATP released by CAAs inhibits the chemotaxis and activity of neutrophils, the final effects in CAA-mediated neutrophil modification on tumor growth and metastasis need further investigations [132].

Adipocytes have been shown to also influence macrophage polarization in the adipose tissue and in the TME. FAs uptake and oxidation were associated with M2 anti-inflammatory, immunosuppressive, and tumor-promoting polarization. In addition, M2 polarization was shown to also be enhanced by adenosine and by CAAs-released lactate through the activation of the ERK/STAT3 pathway [132,137]

Other than neutrophils and macrophages, FAs released by CAA are likely to shut down the anti-tumor immune response, reducing the function of NK and dendritic cells (DCs). In NK cells, robust lipid accumulation in obesity causes complete “paralysis” of their cellular metabolism, decreasing MYC and mTORC1 signals and reducing glycolysis and OXPHOS rates that resulted in less IFN-γ, granzyme B, and perforin production and decreased the cytotoxicity against tumor target cells [138]. In DC, lipids weaken their capability to process tumor antigens and activate T cells effectively [132]. DC activity was shown to also be reduced by CAA-derived extracellular metabolites such as adenosine and lactate that reduce the pro-inflammatory IL-12 and DC antigen presentation and increase the anti-inflammatory IL-10 [132]. In addition, the susceptibility of tumor cells to the NK cytotoxic action was found to be reduced by CAA-derived IL-6 and leptin that activate the JAK/STAT3 signaling axis in tumor cells [132].

CAAs could increase the tumor immune suppression also facilitating the infiltration of MDSCs and their activity in suppressing T cells. Indeed, leptin was shown to be a strong inductor of tumor accumulation of MDSCs [139], and the availability of FAs in the TME could feed FAO that is necessary for the immunosuppressive activity and tumor promotion of MDSCs [140]. In addition, regulatory T cells (Tregs) that accumulate in certain tumors and suppress antitumor immunity, relying mainly on FA uptake and catabolism, might be positively influenced by CAAs in their differentiation and survival [141]. In addition to FAs, Tregs for their survival and immunosuppressive response benefit from CAAs-released lactate, which activates OXPHOS and adenosine [132]. These metabolic components were also described to directly suppress the cytotoxicity of CD8+ T cells and the differentiation of human effector CD4+ T cells [132].

## 6. Conclusions

Together, the current lines of evidence suggest that in the TME, the very complex scenario of not-mutually-exclusive events, such as the differentiation blockade, dedifferentiation, and trans-differentiation, may guide adipocytes to the CAA phenotype essential for cancer progression. Despite the current understanding of the cell intrinsic and extrinsic factors involved in adipocyte differentiation and dedifferentiation, the core regulatory network that triggers adipocyte dedifferentiation has not been characterized. It is highly likely that the dedifferentiation process is orchestrated by a cascade of complex modulators that also play a role in normal adipogenesis; however, the CAA-activated phenotype suggests the induction of other pathways. Thus far, the data obtained mainly in in vitro co-culture models agreed that the canonical Wnt pathway is a crucial regulator of adipocyte fate. 

Besides tightly dissecting the mechanism of adipocyte dedifferentiation and relevant molecules, several questions remain unanswered and need further research. It is important to understand whether in vivo all adipocytes have an equal ability to undertake dedifferentiation or whether they need to be in an already-altered state; whether cancer cells are all able to interact with adipocytes independently of their intrinsic features; and whether the triggering molecule(s) are the same.

Independently from the causes of CAAs generation, adipokines and oncometabolites such as lactate, FAs, and adenosine released by CAAs, in addition to impacting cancer cell proliferation, invasion, and metastatic potential, are able to profoundly affect the effector functions of immune cells and to shape the immune response. Thus, further studies are needed to explore CAAs as target cells that could support the efficacy of immunotherapeutic treatments and directly/indirectly hamper tumor progression. Moreover, an increased understanding of the signals triggering CAAs holds great promise as a new avenue to treat cancers other than the diseases in which adipocyte dedifferentiation was described (i.e., healing and fibrosis). Transcriptional, metabolic, as well as mechanobiological controls of the tumor-induced CAA phenotype, are all worth further investigation, which will likely offer comprehensive clues for cancer therapies.

## Figures and Tables

**Figure 1 ijms-22-03775-f001:**
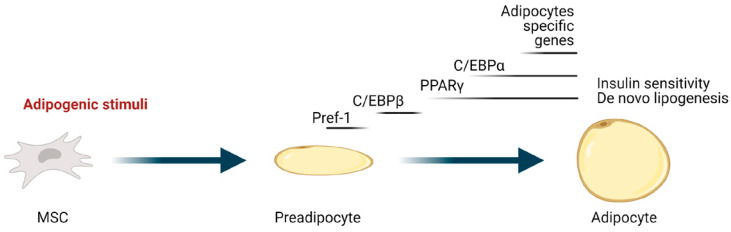
Transcriptional regulators of adipogenesis. Multipotent mesenchymal stem cells (MSC) upon adipogenic stimuli give rise to preadipocytes that, after clonal expansion and subsequent differentiation, become mature adipocytes. Molecules relevant in this process are shown with their approximate induction and duration reflected by lines. The preadipocyte factor 1 (Pref-1) is expressed in preadipocytes and participates in the maintenance of this state. It is an inhibitor of adipogenesis and its expression must be reduced during adipocyte differentiation. CCAAT-enhancer-binding protein C/EBPβ is involved in adipogenesis at an early phase and, together with C/EBPδ, regulates the transcription of the peroxisome proliferator-activated receptor γ (PPARγ). PPARγ with C/EBPα cooperatively induces adipocyte differentiation, regulating the expression of adipocyte-specific genes.

**Figure 2 ijms-22-03775-f002:**
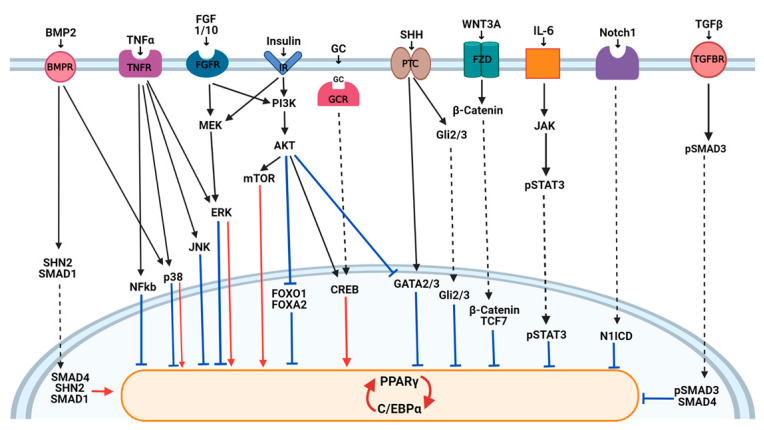
Extracellular regulators of adipogenesis. Signals from activators and repressors of adipogenesis are integrated in the nucleus by transcription factors that directly or indirectly activate (red arrows) or inhibit (blue lines) the expression of peroxisome proliferator-activated receptor γ (PPARγ) and CCAAT-enhancer-binding protein α (C/EBPα). The induction of PPARγ by p38 is related to the BMP2 pathway, and the red arrow from ERK is related to insulin and FGF pathways. AKT, protein kinase B; BMP, bone morphogenetic protein; CREB, cAMP response element-binding protein; ERK, extracellular signal-regulated kinase; FGF, fibroblast growth factor; FGFR, fibroblast growth-factor receptor; FOXO, forkhead protein; FZD, frizzled family receptors; GATA, GATA binding protein; GC, glucocorticoids; GCR, GC receptor; Gli, GLI family zinc finger; IL, interleukin; IR, insulin receptor; JAK, Janus kinase; JNK, Jun *n*-terminal kinase; MEK, mitogen-activated MAPK/ERK kinase; mTOR, mammalian target of rapamycin; N1ICD, Notch1 intracellular domain; NF-kB, nuclear factor kappa-light-chain-enhancer of activated B cells; Notch1; Notch homolog 1, translocation-associated; p38, protein 38 MAPK; PI3K, phosphatidylinositol-3 kinase; PTC, patched; SHN2, schnurri-2; SHH, sonic hedgehog; SMAD, SMAD family member; STAT, signal transducer and activator of transcription; TCF7, T-cell factor 7; TNF, tumor necrosis factor; TNFR, TNF receptor; TGFβ, transforming-growth factor β; WNT, wingless-related integration site.

**Figure 3 ijms-22-03775-f003:**
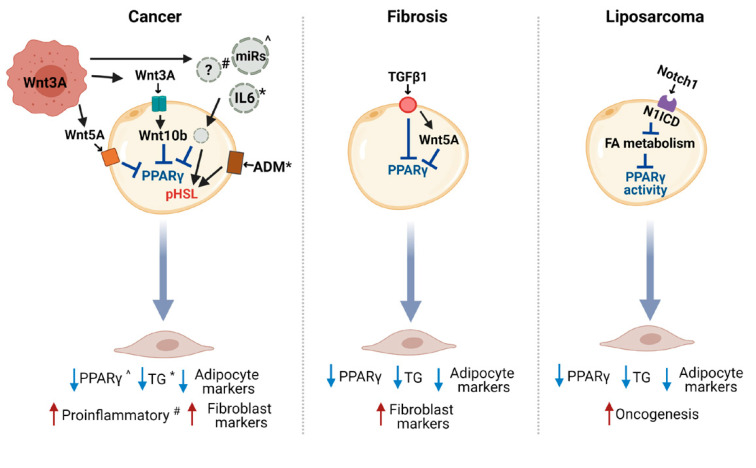
Signaling regulation of adipocyte dedifferentiation. Pathways described to be involved in adipocyte dedifferentiation in cancer and in other pathologies like fibrosis and liposarcoma. ?, indicates that the molecule(s) contained in exosomes and involved in inducing the adipocyte pro-inflammatory phenotype are unknown and have only pro-inflammatory effects (^#^); *, IL-6 was described to induce lipolysis and, thus, to reduce triglyceride (TG) content, while adrenomedullin (ADM) was described to induce the phosphorylation of hormone-sensitive lipase (HSL), a step necessary for lipolysis induction that reduces the TG content; ^, miRs were demonstrated to induce PPARγ reduction. Blue and red arrows indicate reduction and increase, respectively. miR, microRNA; FA, fatty acid; pHSL, phosphorylated hormone sensitive lipase; TGF, transforming growth factor; N1ICD, Notch1 intracellular domain; WNT, wingless-related integration site.

**Figure 4 ijms-22-03775-f004:**
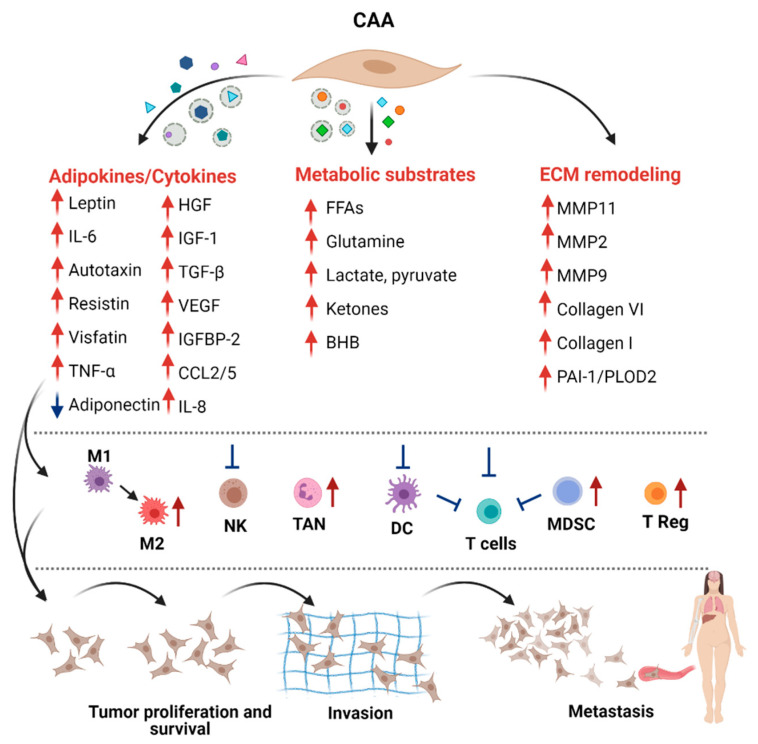
Pro-tumoral factors released by cancer-associated adipocytes (CAAs). CAAs release free or exosome-associated molecules that directly or indirectly, through the modulation of the immune cells, induce the acquisition of aggressive features in breast cancer (BC) cells. The blue and red arrows indicate the increased and decreased release compared to mature adipocytes. In the central panel, blue lines indicate that cells are blocked in their function; red arrows indicate that cells are recruited/activated. BHB, β-hydroxybutyrate; DC, dendritic cells; FFAs, free fatty acids; IL, interleukin; HGF, hepatocyte growth factor; IGF, insulin growth factor; IGFBP2, IGF binding protein 2; M, macrophages; MDSC, myeloid derived suppressor cells; NK, natural killer cells; PAI-1, plasminogen activator inhibitor 1; PLOD2, procollagen-lysine 2-oxoglutarate 5-dioxygenase 2; TGFβ, transforming growth factor β; TAN, tumor associated neutrophils; TNF, tumor necrosis factor; T Reg, regulatory T cells; VEGF, vascular endothelial growth factor.

## Data Availability

Not applicable.

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
