# Peer review of "Cancer-Associated Adipocytes in Breast Cancer: Causes and Consequences"

_ijms, 2021, doi:10.3390/ijms22073775_

Round 1
Reviewer 1 Report
This article presented by Rybinska et al. is reviewing the role of cancer associated adipocytes (CAA) in breast cancer. The authors highlight several important topics, such as regulation of adipocyte differentiation and CAA induction and they provide a comprehensive overview about those topics, with a particular focus on the interplay of the pathways involved.
Understanding the molecular processes underlying the development of adipocytes and the processes leading to induction of CAAs is important for the field and for developing novel treatment strategies.
However, there are a number of shortcomings currently dampening enthusiasm to see this article published in its current state as outlined below. If the authors address these issues, the article may provide a valuable overview about recent developments and topics discussed therein.
Conceptual issues:
Some of the authors published recently another review article on a related topic, although with a slightly different focus (doi: 10.3390/cells9030560). Therefore, the authors may consider extending or adding some sections such as discussing recent developments in therapeutic approaches, if possible. This could increase the value for the scientific community and its exclusivity.
Textual issues:
The article is generally well written and easy to follow. There are however some textual issues that should be addressed:
- The authors should cite the original research paper, not a review article when describing scientific results in detail. For example, line 204, reference 41 or line 222, 224 reference 43,44 in which the exact reference is not quite clear.
- Reference 4 does not mention adipocytes and thus the citation is misleading (line 34).
- Reference 1 is referring to a statistic article from 2019 that is updated every year. Therefore, I suggest citing the most recent version of this article (2021). Further, this article is referring to cancer statistics from the US, but does not describe in detail the situation worldwide. The authors may want to cite an additional article referring to the situation worldwide or rephrase the sentence accordingly (line 23).
- The authors report that "...there is growing evidence that support the interaction between adipocytes and cancer cells..." (line 38-40). The article cited is a review article (reference 7) and it should be labelled as such.
- There are several examples in this article where it is not quite clear if the experiments the author discuss are in vitro, in vivo, human or mouse based. I think communicating this in a clearer fashion helps to understand the current state of research in this field and thus it should be addressed. Some examples: line 83, 97, 166, 236.
- 3T3-L1 cells are introduced as murine (preadipocyte?) cells in line 128 but the name/term is used before. As this cell line appears to be central in this field, I’d suggest introducing them earlier (e.g., line 84).
- The 3T3-L1 cells appear to be introduced as murine preadipocyte cells (line 128, 129), but later in this article described as adipocytes, line 292. This is confounding and can be phrased more precisely.
- Line 355. The authors may rephrase this “…that have activation of Notch1 signaling specifically…” to e.g. "…that over-express the activated form of the Notch1 receptor (NICD)…”
- Line 364 “…in the cells caused by…”. It appears that there is a linking verb missing, such as “is”. This sentence may be rephrased to make it easier to understand.
- Line 487-491. This sentence should be rephrased and shortened.
- Line 141. The authors should add information about the protease leading to liberation of the Notch-ICD. Otherwise, mentioning the use of y-secretase inhibitor (line 144) is difficult to understand.
- Figure 1A. It is conceptually unclear what the right panel (black lining) is supposed to express. One can guess from the manuscript text, but it should be clear at a glance or described in greater detail in the figure legend.
- Figure 1B. This figure appears to be too busy; the arrows and text are difficult to read and the pathway symbols (outer circle layer) are not evenly distributed (in case that was the aim). I suggest to re-work parts of this figure to enhance readability.
Author Response
We thank the reviewer for his/her positive comments and suggestions.
Comment: Some of the authors published recently another review article on a related topic, although with a slightly different focus (doi: 10.3390/cells9030560). Therefore, the authors may consider extending or adding some sections such as discussing recent developments in therapeutic approaches, if possible. This could increase the value for the scientific community and its exclusivity.
Response: We believe that the scope and the content of our previous review are much different from this one. In our previous review article (doi: 10.3390/cells9030560) we were principally focused on protumoral phenotype of cancer associated adipocytes (CAAs) and we compared them to adipocytes in other tumor-related pathologies such as obesity and cancer cachexia, emphasizing the substantial similarities in their phenotypes. Interestingly, reduced adipogenesis is a defining characteristic of CAAs and adipocytes in obesity and cancer cachexia, but there is still lack of knowledge regarding triggers regulating that process in cancer. Although the current paper includes also description of CAA phenotype and their role in BC progression, we are mainly focused on factors and mechanisms regulating adipogenesis and inducing adipocyte dedifferentiation.
Comment: The authors should cite the original research paper, not a review article when describing scientific results in detail. For example, line 204, reference 41 or line 222, 224 reference 43,44 in which the exact reference is not quite clear.
Response: We thank the reviewer for this comment. We changed the references accordingly.
Line 223 (ref 49), line 238 (ref 54, 55), line 266(ref 49, 62), line 267 (ref 63), line 285 (ref 66, 68), line 288 (ref 69, 70), line 290 (ref 71).
Comment: Reference 4 does not mention adipocytes and thus the citation is misleading (line 34).
Response: We changed the reference into: Song T, Kuang S. Adipocyte dedifferentiation in health and diseases. Clin Sci (Lond). 2019;133(20):2107-2119. doi:10.1042/CS20190128 (ref 4).
Comment: Reference 1 is referring to a statistic article from 2019 that is updated every year. Therefore, I suggest citing the most recent version of this article (2021). Further, this article is referring to cancer statistics from the US, but does not describe in detail the situation worldwide. The authors may want to cite an additional article referring to the situation worldwide or rephrase the sentence accordingly (line 23).
Response: We thank the reviewer for this suggestion. We changed the reference inserting the one suggested by reviewer 2 (GLOBOCAN 2020) that referred to cancer statistics worldwide.
Comment: The authors report that "...there is growing evidence that support the interaction between adipocytes and cancer cells..." (line 38-40). The article cited is a review article (reference 7) and it should be labelled as such.
Response: We changed the labeling of all references that are review articles as suggested:
Line 41, 51, 137, 162, 205, 212, 234,255.
Comment: There are several examples in this article where it is not quite clear if the experiments the author discuss are in vitro, in vivo, human or mouse based. I think communicating this in a clearer fashion helps to understand the current state of research in this field and thus it should be addressed. Some examples: line 83, 97, 166, 236.
Response: We thank the reviewer for this comment. We added the requested information:
Line 163, 171,200,221,224,227, 236, 264, 273, 282, 286, 290, 295, 359, 365, 397, 405, 422, 503.
Comment: 3T3-L1 cells are introduced as murine (preadipocyte?) cells in line 128 but the name/term is used before. As this cell line appears to be central in this field, I’d suggest introducing them earlier (e.g., line 84).
The 3T3-L1 cells appear to be introduced as murine preadipocyte cells (line 128, 129), but later in this article described as adipocytes, line 292. This is confounding and can be phrased more precisely.
Response: To avoid confusion, we insert a sentence in line 62 to describe 3T3-F442A and 3T3-L1 cells used in fat metabolism studies. “Adipocytes derived from differentiation of murine 3T3-F442A or 3T3-L1 cells that are clonal sublines isolated from 3T3 mouse embryonic fibroblasts, undergo sequential phenotypic and functional alterations that make them different from mature adipocytes from which they derive upon exposure to cancer cells or their conditioned media in vitro”. They are considered preadipocytes and they are able to differentiate in adipocytes. Thus, 3T3-L1 adipocytes means that are adipocytes derived from 3T3-L1 cells.
Comment: Line 355. The authors may rephrase this “…that have activation of Notch1 signaling specifically…” to e.g. "…that over-express the activated form of the Notch1 receptor (NICD)…”
Response: The sentence in line 355 (new line 429) has been changed as suggested: “The analysis of Ad/N1ICD mice that overexpress the activated form of the Notch1 receptor (NICD) specifically in mature adipocytes showed that they have significantly lower body fat mass compared to controls and gene expression analysis in the adipose tissue depots showed a significant repression of lipogenic and adipogenic pathways (C/EBPα, C/EBPβ, SREBF1).”
Comment: Line 364 “…in the cells caused by…”. It appears that there is a linking verb missing, such as “is”. This sentence may be rephrased to make it easier to understand.
Response: We rephrased the sentence into: ”Since PPARγ was overexpressed in tumor cells, it is likely that dedifferentiation in Ad/N1ICD adipocytes derives from deficient activation of PPARγ due to the suppression in the cells of lipid metabolism pathways that supply ligands to PPARy” as suggested (line 437).
Comment: Line 487-491. This sentence should be rephrased and shortened.
Response: We rephrased the sentence into: ” FFAs uptake in cancer cells is mediated by several proteins including fatty acid translocase (FAT)/CD36, fatty acid transport proteins (FATPs), low density lipoprotein receptor (LDLR), and fatty acid binding proteins (FABPs) (reviewed in [117]). The uptake of FFAs by CD36 was described to increase BC cell growth in vitro [118] and its expression in human BC samples was associated with poor prognosis [119].” as suggested (line 565).
Comment: Line 141. The authors should add information about the protease leading to liberation of the Notch-ICD. Otherwise, mentioning the use of y-secretase inhibitor (line 144) is difficult to understand.
Response: We add the requested information (line 198): ”Binding of ligands to Notch induced proteolytic cleavage of Notch by the γ-secretases and translocation of its intracellular domain (N1ICD) to the nucleus”.
Comment: Figure 1A. It is conceptually unclear what the right panel (black lining) is supposed to express. One can guess from the manuscript text, but it should be clear at a glance or described in greater detail in the figure legend.
Figure 1B. This figure appears to be too busy; the arrows and text are difficult to read and the pathway symbols (outer circle layer) are not evenly distributed (in case that was the aim). I suggest to re-work parts of this figure to enhance readability.
Response: We decided to separate the panels in two figures to make panel B more readable and panel A more clear (figura 1A to figure 1, figure 1B to figure 2). Black lining in figure 1 wants to depict the timing of the expression of crucial transcription factors and genes during the cascade of adipocyte differentiation. Details have been added in figure legend (lines 120-126). Figure 2 has been changed to make the figure more readable.

Reviewer 2 Report
Dear Authors,
congratulations on such interesting and very well described paper. I have only several minor suggestions:
- Line 24 - please replace the reference with the most recent one - GLOBOCAN 2020. It is the most updated version regarding cancer statistics.
- Line 27 - TME rather does not affect but constitutes a major part involved in the possible modulation of processes associated with the hallmarks of carcinogenesis. Please correct.
- I would recommend replace the paragraph number 2 with three - just the order of the paragraphs in the text. It will be clearer for the readers to read.
best regards with your further research!
A reviewer
Author Response
We thank the reviewer for his positive comment.
Comment: Line 24 - please replace the reference with the most recent one - GLOBOCAN 2020. It is the most updated version regarding cancer statistics.
Response: We thank the reviewer for this suggestion. We have changed the reference n°1 (line 24).
Comment: Line 27 - TME rather does not affect but constitutes a major part involved in the possible modulation of processes associated with the hallmarks of carcinogenesis. Please correct.
Response: We have changed the sentence as requested into: ” The TME indeed actively contributes to the acquisition of cancer hallmark traits like angiogenesis, epithelial to mesenchymal transition (EMT), proliferation, invasion and metastasis” (line 27).
Comment: I would recommend replace the paragraph number 2 with three - just the order of the paragraphs in the text. It will be clearer for the readers to read.
Response: We thank the reviewer for this suggestion. We changed the order of the two paragraphs as suggested.
